# A High-Performance Liquid Chromatography—Mass Spectrometry Method for Simultaneous Determination of Vancomycin, Meropenem, and Valproate in Patients with Post-Craniotomy Infection

**DOI:** 10.3390/molecules28062439

**Published:** 2023-03-07

**Authors:** Yuting Jin, Qiang Sun, Yumei Pei, Jing Huang

**Affiliations:** Department of Laboratory Medicine, The First Hospital of Jilin University, Changchun 130021, China

**Keywords:** cranial infection, vancomycin, meropenem, valproate, LC–MS/MS

## Abstract

Vancomycin (VAN), meropenem (MER), and valproate (VPA) are commonly used to treat intracranial infection post-craniotomy and prevent associated epilepsy. To monitor their levels, we developed a novel bioassay based on liquid chromatography–tandem mass spectrometry (LC–MS/MS) for simultaneous determination of these three drugs in human serum and cerebrospinal fluid (CSF). Sample preparation by protein precipitation using acetonitrile was followed by HPLC on a Zorbax 300SB-C8 column (150 mm × 4.6 mm, 5 μm) maintained at 40 °C. The lower limit of quantification (LLOQ) was 5 ng/mL for MER, 0.1 μg/mL for VAN, and 1 μg/mL for VPA in serum and 50 ng/mL for MER, 1 μg/mL for VAN, and 2 μg/mL for VPA in CSF. This method was validated with satisfactory linearity, sensitivity, precision, accuracy, recovery, matrix effects, and stability for all analytes. The assay was then successfully applied to evaluate VPA, MER, and VAN levels in serum and CSF from patients with intracranial infection administrated by intrathecal injection. Compared with intravenous injections, an intrathecal injection can provide sufficient therapeutic effects even if the CSF levels did not reach the effective concentration reported. Our method provided a detection tool to study the effective concentrations of these three drugs in CSF from patients administered via intrathecal injection.

## 1. Introduction

Central nervous system infection is a common and serious complication after neurosurgery that affects prognosis and outcomes and may become life-threatening [1,2]. VAN and MER are the first-line drugs for the treatment of medical-related central nervous system infections [3,4]. Brain diseases can cause nerve cell damage, abnormal discharge, and increase the risk of secondary seizures [5]. As a first-line treatment for various types of epilepsy, VPA is also widely used to prevent seizures in neurosurgery patients [6]. The combination of these three drugs appears to be a reasonable treatment option. However, these three drugs are associated with high toxicity and high risks of adverse reactions [7,8,9]. Therefore, therapeutic drug monitoring is mandatory for the combination of VAN, MER, and VPA.

Antibiotic drugs rarely pass through the blood-brain barrier (BBB), which makes anti-infective therapy after neurosurgery challenging [10]. In addition, craniocerebralinjury may lead to the opening of the BBB and the degree of leaking varies with different pathological conditions, leading to varied drug absorption from blood to CSF. In recent years, the administration of broad-spectrum antibiotics by intrathecal injection has been used to treat an intracranial infection. Compared with intravenous injection, an intrathecal injection can greatly increase drug concentration in CSF and offers superior safety and therapeutic effects [11]. Thus, concentrations of these three drugs in serum and CSF are potential quantitative traits that can describe and define each patient’s therapeutic characteristics. However, the analytical methods available to determine the levels of these three drugs with high selectivity and sensitivity are currently limited.

The multiple reaction monitoring (MRM) method using LC–MS/MS is a common analytical method that can detect various compounds with high sensitivity and specificity. Detection of VAN and MER in human plasma with an LLOQ of 1 μg/mL, extracted by protein precipitation, was reported by Barco et al. [12,13]. Lipska et al. achieved a much lower LLOQ of 0.075 μg/mL for VPA in plasma using high-performance liquid chromatography with ultra-violet detection (HPLC-UV) and gas chromatography–mass spectrometry (GC-MS) methods [14]. Ye et al. developed a high-throughput LC–MS/MS approach to measuring VAN in human CSF with an LLOQ of 0.1 μg/mL [15]. Lu et al. developed an LC–MS/MS approach to detect 18 antibacterial drugs in human plasma with an LLOQ of 2.02 μg/mL for MER and 0.41 μg/mL for VAN [16]. However, there are still no analytical methods available to determine MER in CSF, and the reported methods for MER in plasma or serum always possess higher LLOQ. Although these three drugs are used together in clinical practice, there is no analytical method available for detecting them in plasma, serum, or CSF simultaneously.

In this work, we present a novel LC–MS/MS method for the simultaneous detection of VAN, MER, and VPA in human serum and CSF. This assay was then applied to patients with postoperative intracranial infection treated with VAN, MER, and/or VPA via intrathecal injection. Our method provided a detection tool to study the effective concentrations of these three drugs in CSF from patients administered via intrathecal injection.

## 2. Results and Discussion

### 2.1. LC–MS/MS Conditions

Due to their strong hydrophily, VAN and MER are not well retained in the C18 column; thus, we selected the C8 column to improve retention efficiency. Acetonitrile and methanol were tested for the mobile phase. Acetonitrile was chosen for the mobile phase because it provided lower background interferences and column pressure than methanol. To evaluate the effects of different additives on the mobile phase, we compared the addition of ammonium acetate, formic acid (FA), and acetic acid at different concentration levels. The addition of ammonium acetate can increase the buffering capacity of the system but leads to a decreased analyte signal. The addition of acetic acid could not improve the signal intensities of analytes at 0.05% and improved slightly at 0.1%. Formic acid (FA) was added to the mobile phase to improve the ionization of the analytes, and 0.05% FA in the mobile phase instead of 0% and 0.1% were used to improve the signal intensities and peak shapes. With the appropriate mobile phase (0.05% FA), the analytes showed good chromatographic peak shapes, minimum interference, and stable retention times.

VPA in the positive ion mode was unsuitable due to strong interference from endogenous compounds; it could only be detected in the negative ion mode. Therefore, MS should be switched between positive and negative ion modes for the simultaneous determination of these three analytes.

### 2.2. Sample Preparation

Due to significant differences in the lipophilicity of these analytes, protein precipitation was selected for sample preparation. During the optimization of the sample preparation step, methanol and acetonitrile were tested for protein precipitation. However, the analyte peaks were strongly influenced when protein-precipitated by methanol. It is possibly caused by the different lipophilicity of mobile phase and precipitation solvents, leading to a solvent effect. However, the calibration curves of VPA in samples precipitated by acetonitrile alone exhibited non-linearity. We overcame this issue by adding a mixture of acetonitrile and acid to precipitate the protein. The addition of different FA and acetic acid products different r values for the calibration curves of VPA: r = 0.9659 for 0.2% FA; r = 0.9960 for 0.5% FA; and r = 0.9689 for 1% FA; r = 0.9539 for 0.2% acetic acid; r = 0.9605 for 0.5% acetic acid; r = 0.9830 for 1% acetic acid. In this study, 0.5% FA in acetonitrile was selected for protein precipitation. Notably, the stability of MER in serum and plasma differed at RT, with MER in serum showing greater stability (Figure 1). Therefore, a serum sample is chosen for clinical detection in this study.

### 2.3. Assay Performance and Validation

The method was confirmed to be specific, as no peaks due to endogenous components were observed in serum or CSF around the retention times of t he analytes and IS (Figure 2). In addition, no crosstalk or carryover was observed among analytes and IS. The calibration curves were linear over the range 5–500 ng/mL for MER, 0.1–10 μg/mL for VAN and 1–100 μg/mL for VPA in serum and 50–5000 ng/mL for MER, 1–100 μg/mL for VAN and 2–200 μg/mL for VPA in CSF with the following equations: y = 0.0044x + 0.00033, r = 0.9970 for MER in serum; y = 0.0109 × − 0.00023, r = 0.9984 for VAN in serum; y = 0.0195 x − 0.002, r = 0.9955 for VPA in serum; y = 0.0006x + 0.00198, r = 0.9959 for MER in CSF; y = 0.0805x − 0.00944, r = 0.9982 for VAN in CSF; and y = 0.0108x + 0.00087, r = 0.9947 for VPA in CSF. The LLOQs of this method were 5 ng/mL for MER, 0.1 μg/mL for VAN, and 1 μg/mL for VPA in serum and 50 ng/mL for MER, 1 μg/mL for VAN and 2 μg/mL for VPA in CSF with signal-to-noise ratio > 10:1 (Figure 2), and accuracies and precisions within 11.1% (Table 1). 

Table 1 presents the accuracy and precision data for each analyte in serum and CSF. The intra- and inter-day accuracies (as relative error, RE) ranged from −6.2 to 6.3% with precision (as relative standard deviation, RSD) ≤ 7.4% (intra-day) and 14.1% (inter-day), respectively, for each analyte in serum and CSF.

The matrix effects and recovery efficiency results showed that the recoveries were all reproducible in low, medium, and high QC samples and consistent across the concentration range studied. In terms of matrix effects, Table 2 shows the actual concentrations (mean ± SD) as the percentage of nominal concentration. Matrix effects of VAN, MER, and VPA in serum and CSF were low based on Table 2. Therefore, the detection of these three drugs in serum and CSF were not affected by the sample from different patients.

In terms of stability (Table 3), the concentrations under the various test conditions were all within ±12.8% of the nominal concentrations, indicating no significant degradation of the analytes under any of the storage conditions tested. Stock solutions of MER, VAN, and VPA in acetonitrile: water (1:1, v:v) are stable at −20 °C for 1 month and stable at RT for 24 h with a range from 94.1 to 107.7% and RSD% ≤ 5.44%. The evaluation of MER concentration in six 5000 ng/mL of serum samples and 50,000 ng/mL of CSF samples after 10-fold dilution with blank serum and CSF resulted in an accuracy of 101.6% and RSD of 3.8% for serum and accuracy of 97.4% and RSD of 4.1% for CSF. Thus, the developed assay can be used to quantify MER in human serum and CSF samples that are diluted to a maximum of 10 folds.

### 2.4. Clinical Application

Based on prior studies [17], the CSF to serum albumin ratio (CSAR; calculated as CSF albumin (mg/L)/serum albumin (g/L)) was used to evaluate BBB function. The definition of BBB damage was derived from age-adjusted reibergrams (normal if < 6.5 in patients aged < 40 years or < 8 in patients > 40 years). All patients selected in this study experienced intracranial infection after craniotomy and suffered from varying degrees of BBB damage. The effective concentrations of MER, VAN, and VPA in plasma were 4–10 μg/mL [18], 5–10 μg/mL [19], and 50–100 μg/mL [20], respectively. The CSAR and drug concentrations for the selected patients in this study are shown in Table 4. The developed LC–MS/MS method was able to capture all the serum and CSF concentrations of VAN and VPA, and MER with 10-fold dilution in all patients.

In this study, due to administrate via intrathecal injection CSF concentrations were higher than serum concentrations. Compared with intravenous injections, an intrathecal injection can offer superior therapeutic effects. Although the CSF levels did not reach the effective concentration reported, these three drugs still provide sufficient therapeutic effects because the effective concentrations reported were obtained from clinical studies administrated via intravenous injections. When administrated by intravenous injection, the actual concentrations in CSF were significantly lower than in serum. It indicated the effective concentrations of these three drugs in CSF administrated by intrathecal injection may be lower than the reported effective concentrations based on the clinical study administrated via intravenous injections. Our method provided a detection tool to study the effective concentrations of these three drugs in CSF from patients administered via intrathecal injection. To prevent and reduce adverse drug reactions while ensuring therapeutic effects, the effective concentrations of these three drugs in CSF administrated by intrathecal injections are highly essential. In addition, even when the same dose is administered, the drug concentration varies greatly across different patients. Such variation is affected by many factors such as the physiological and pathological states [21,22,23]. For this reason, it is necessary to monitor drug concentrations in CSF and serum.

## 3. Materials and Methods

### 3.1. Chemicals and Reagents

VAN (purity > 95%), MER (purity > 86%), and VPA (purity > 99.5%) were purchased from the National Institute for Food and Drug Control (Beijing, China). VPA-^2^H_6_ (VPA-D6, purity > 99.4%) and MER-D6 (purity > 99.8%), which were used as internal standards (IS), were purchased from Sigma-Aldrich (St. Louis, MO, USA). HPLC grade acetonitrile was obtained from Fisher Scientific (Fair Lawn, NJ, USA). FA was obtained from the Beijing Chemical Plant (Beijing, China). Purified water was produced using a Milli-Q system (Millipore, Bedford, MA, USA).

### 3.2. LC Conditions

Separation was performed on Zorbax Eclipse XDB-C8 columns (150 mm × 4.6 mm, 5 μm) maintained at 40 °C. The mobile phase consisted of 0.05% formic acid in water (solvent A) and 0.05% formic acid in acetonitrile (solvent B) delivered at a flow rate of 0.8 mL/min. The gradient elution program was as follows: 0–2 min, 10% B; 2–2.5 min, 10–20% B; 2.5–3 min, 20–75% B; 3–6 min, 75% B; 6–6.1 min, 75–10% B. 6.1–8 min, 10% B.

### 3.3. MS Conditions

Detection employed a Qtrap 5500 mass spectrometer (AB Sciex, Concord, ON, Canada) equipped with a TurboIon Spray™ source. Data acquisition and integration were performed using Applied Biosystems Analyst software version 1.6.3. MS parameters optimized for each analyte and IS by infusion of their respective standards using a syringe pump at a flow rate of 20 μL/min. The declustering potentials (V) and collision energies (eV) were as follows: VAN 100, 20; MER 140, 33; VPA −180, −10; MER-D6 110, 21, and VPA-D6 −190, −12. VAN, MER, and MER-D6 were detected by positive ion ESI and the transitions (*m*/*z*) for MRM were VAN 725.5 → 144.2, MER 384.3 → 113.9, and MER-D6 390.3 → 147.3. VPA and VPA-D6 were detected by negative ion ESI and the transitions (*m*/*z*) for MRM were VPA 143.3→143.3 and VPA-D6 149.3→149.3 (Figure 3).

### 3.4. Preparation of Calibration Standards and QC Samples

All solutions were prepared in acetonitrile: water (1:1, *v*:*v*). Blank human serum and CSF were prepared by mixing equivolumes from six different patients without administration of these three drugs. Stock solutions of the analytes (10 mg/mL) were diluted to achieve standard solutions for serum with concentrations of 50, 100, 300, 500, 1000, 3000, and 5000 ng/mL for MER; 1, 2, 6, 10, 20, 60, and 100 μg/mL for VAN; and 10, 20, 60, 100, 200, 600, and 1000 μg/mL for VPA. Calibration standards were prepared by diluting standard solutions with blank serum to yield concentrations of 5, 10, 30, 50, 100, 300, and 500 ng/mL for MER; 0.1, 0.2, 0.6, 1, 2, 6, and 10 μg/mL for VAN; and 1, 2, 6, 10, 20, 60, and 100 μg/mL for VPA. QC samples with concentrations of 8, 75, and 400 ng/mL for MER; 0.16, 1.5, and 8 μg/mL for VAN; and 3.2, 30, and 160 μg/mL for VPA were prepared in a similar way. The IS stock solution (1 mg/mL) was diluted to produce IS working solutions (20 μg/mL for VPA-D6 and 2 μg/mL for MER-D6).

Stock solutions of analytes (10 mg/mL) were diluted by acetonitrile: water (1:1, *v*:*v*) to give standard solutions for CSF with concentrations of 0.5, 1, 3, 5, 10, 30, and 50 μg/mL for MER; 10, 20, 60, 100, 200, 600, and 1000 μg/mL for VAN; and 20, 40, 120, 200, 400, 1200, and 2000 μg/mL for VPA. Calibration standards were prepared by diluting standard solutions with blank CSF to give concentrations of 50, 100, 300, 500, 1000, 3000, and 5000 ng/mL for MER; 1, 2, 6, 10, 20, 60, and 100 μg/mL for VAN; and 2, 4, 12, 20, 40, 120, and200 μg/mL for VPA. QC samples with concentrations of 80, 750, and 4000 ng/mL for MER; 1.6, 15, and 80 μg/mL for VAN; and 3.2, 30, and 160 μg/mL for VPA were prepared in a similar way. The IS stock solution (1 mg/mL) was diluted to produce IS working solutions (40 μg/mL for VPA-D6 and 1 μg/mL for MER-D6). Stock and standard solutions were stored at −20 °C when not in use.

### 3.5. Sample Preparation

A 50 μL sample (or calibration standard or QC sample) was added to 50 μL IS working solution and 400 μL 0.5% FA in acetonitrile. After vortexing for 30 s and centrifuging at 9000× *g* for 10 min, the supernatant (5 μL) was injected into the LC–MS/MS system. A matrix comparison between human serum and plasma (EDTA) was also performed to investigate the stability of the assay in human matrices (Appendix A).

### 3.6. Assay Validation

Assay validation was performed in accordance with the US FDA Guidance for Industry on Bioanalytical Method Validation [24]. Specificity was determined by analysis of blank human serum and CSF samples from six different individuals. Linearity was evaluated by fewest squares weighted (1/×2) linear regression of duplicate calibration curves prepared using three separate batches of serum and CSF. The lowest concentration on the calibration curve that could be determined with 80.0–120.0% accuracy and had RSDs not exceeding 20.0% was chosen as the LLOQ. Accuracy and precision were determined by analyzing QC samples on three separate days. Carryover was assessed by analysis of blank serum and CSF immediately following injection of an upper limit of quantitation (ULOQ) sample. Carryover was considered negligible if the peak area at the retention time of the analyte in the blank sample was <20% of the analyte peak area in the LLOQ sample. Recovery was determined by comparing the mean peak areas of analytes in six replicate QC samples with those of post-extraction blank serum and CSF samples spiked at the same concentrations. Matrix effects were determined by comparing the mean peak areas of post-extraction blank serum and CSF samples spiked at QC concentrations with those of corresponding standard solutions. Stability in serum and cerebrospinal fluid was assessed by assay of triplicate QC samples after storage at −20 °C for 30 days, at room temperature (RT) for 6 h, and after three freeze–thaw cycles (−20 °C to RT). Stability in processed samples was assessed by assay of triplicate QC samples after storage in the autosampler at 15 °C for 6 h and at RT for 6 h. The stability of stock solution in acetonitrile: water (1:1, *v*:*v*) was evaluated at −20 °C for 1 months and at RT for 24 h. MER, VAN, and VPA with final concentrations of 100 ng/mL, 2 μg/mL and 20 μg/mL in mobile phase were prepared in triplicate from old and freshly made stock solutions. The stability of stock solution was tested by comparing the instrument response of samples prepared from old stock solution with that of samples prepared from new stock solution. To evaluate dilution integrity of MER samples, serum and CSF samples with final MER concentrations of 5000 ng/mL and 50,000 ng/mL (10 times ULOQ) were prepared in six replicates and diluted with blank serum and CSF to give the final concentration of 500 ng/mL and 5000 ng/mL (10-fold dilution). Diluted samples were processed following the method described in Section 3.5. and analyzed with calibration standards prepared on the same day. Accuracy and precision within ±15.0% were set as acceptance criteria.

### 3.7. Clinical Assessment

To demonstrate the clinical value of the proposed assay, 62 plasma and CSF samples obtained from patients (age range: 16−69 years old) at the First Hospital of Jilin University in 2022 were analyzed. All enrolled patients had experienced intracranial infection and were treated with VAN, MER, and/or VPA at 0.5 g, 2 g, and 1.5 g via intrathecal injection after craniotomy. This study was approved by the Human Research Ethics Committee of the First Hospital, Jilin University (protocol code is 2022-037 and date of approval is 22 February 2022).

Samples were analyzed to quantify MER, VPA, and VAN. When the level of MER in the serum or CSF samples was over 0.5 µg/mL or 5 µg/mL, the samples were 10-fold diluted by blank serum or CSF. Serum samples were obtained after centrifugation of blood in coagulation-promoting vacuum tubes. Isolated serum and CSF were immediately frozen (–20 °C) until sample preparation. To observe the stability of MER at RT, we used samples from six healthy subjects whose blood was collected with K2-EDTA vacuum tubes and coagulation-promoting vacuum tubes to obtain plasma and serum samples after centrifugation, respectively.

## 4. Conclusions

VAN, MER, and VPA are commonly used to treat intracranial infection after craniotomy and to prevent epilepsy. Monitoring these drugs in the patients’ serum and CSF could be used to adjust the dosages. This approach could help realize individualized medication and prevent and reduce drug adverse reactions while ensuring the therapeutic effect. In this study, an LC–MS/MS method for simultaneous determination of MER, VAN, and VPA in CSF and serum was developed and validated. The assay has several favorable characteristics including rapid sample preparation, a small sample volume of 50 μL, and a short cycle time (8 min), allowing a high-throughput analysis for clinical therapeutic drug monitoring. Although the CSF levels in this study did not reach the effective concentration reported, these three drugs still provide sufficient therapeutic effects. It indicated the effective concentrations of these three drugs in CSF administrated by intrathecal injection may be lower than those administrated via intravenous injections. It is very essential to investigate the effective concentrations of these three drugs in CSF administrated by intrathecal injection. Our method provided a detection tool to investigate the effective concentrations of these three drugs in CSF from patients administered via intrathecal injection. For this reason, our method is an effective analysis approach for clinical purposes.

## Figures and Tables

**Figure 1 molecules-28-02439-f001:**
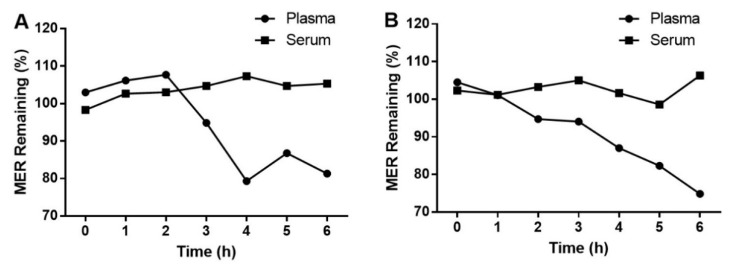
Stability of MER in serum and plasma at LQC (**A**) and HQC (**B**) at RT.

**Figure 2 molecules-28-02439-f002:**
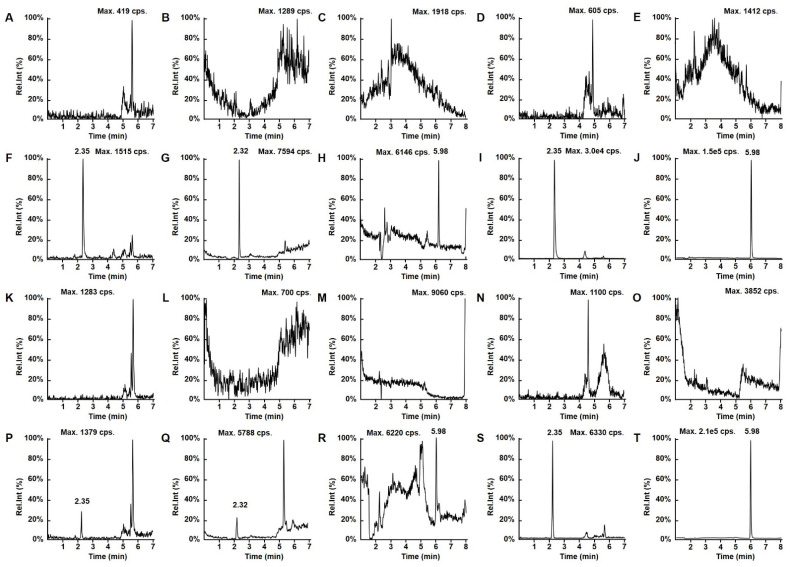
Representative LC–MS/MS chromatograms for (**A**) MER, (**B**) VAN, (**C**) VPA, (**D**) MER-D6, and (**E**) VPA-D6 in blank CSF; (**F**) MER, (**G**) VAN, (**H**) VPA, (**I**) MER-D6, and (**J**) VPA-D6 in CSF at LLOQ; (**K**) MER, (**L**) VAN, (**M**) VPA (**N**) MER-D6, and (**O**) VAN-D6 in blank serum; (**P**) MER, (**Q**) VAN, (**R**) VPA (**S**) MER-D6, and (**T**) VAN-D6 in serum at LLOQ.

**Figure 3 molecules-28-02439-f003:**
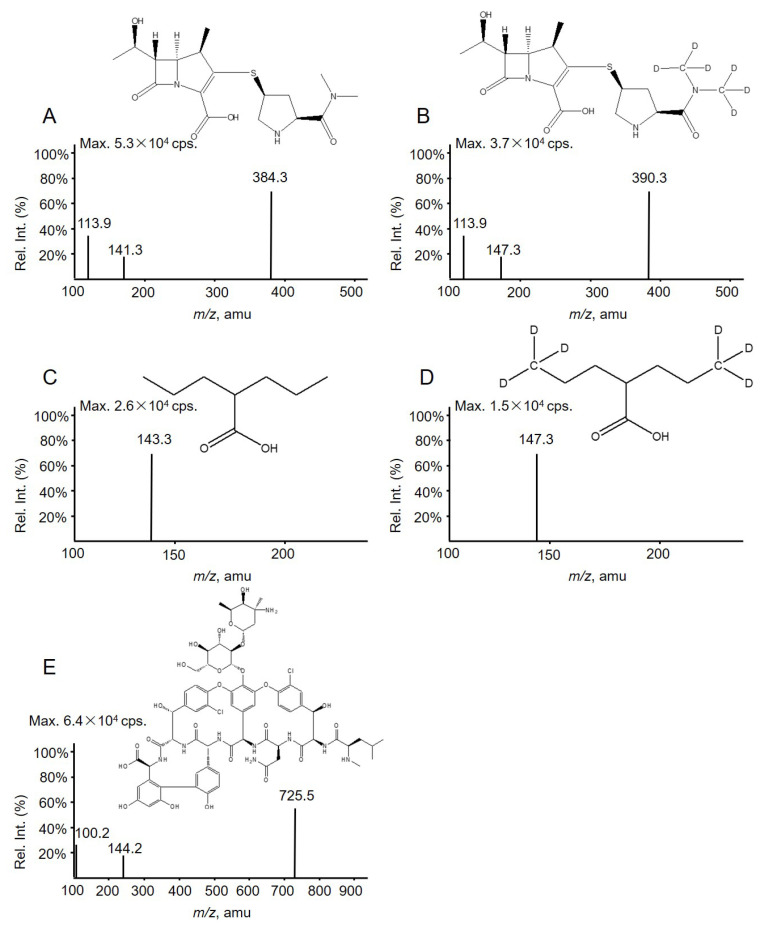
Full-scan product ion spectra of (**A**) MER, (**B**) MER-D6 (IS), (**C**) VPA, (**D**) VPA-D6 (IS), and (**E**) VAN along with their structure.

**Table 1 molecules-28-02439-t001:** Accuracy and precision for determination of VAN, MER, and VPA in serum and CSF (data based on assays of five replicates on three different days).

Analyte	Matrix	Concentration (μg/mL)		Precision (RSD %)		Accuracy (RE %)
		Nominal	Mean Found	Intra-Day	Inter-Day	
VAN	Serum	0.1	0.11	6.5	3.6	5.6
		0.16	0.16	4.6	4.4	−1.0
		1.5	1.45	4.8	4.0	−3.0
		8	7.98	3.1	5.7	−0.2
MER		0.005	0.005	5.3	5.5	2.9
		0.008	0.008	5.7	8.3	3.9
		0.075	0.075	3.8	4.0	−6.1
		0.400	0.387	2.0	3.7	−3.3
VPA		1	1.06	4.0	2.0	6.3
		1.6	1.66	3.0	4.5	3.5
		15	14.46	4.2	13.3	−3.6
		80	82.42	3.7	9.3	3.0
VAN	CSF	1	1.06	2.9	7.6	5.3
		1.6	1.69	4.7	6.7	5.5
		15	15.19	5.6	7.2	1.3
		80	77.63	1.1	7.5	−3.0
MER		0.050	0.050	7.4	6.4	0.8
		0.080	0.080	3.0	13.9	−0.1
		0.750	0.745	2.3	11.1	−0.7
		4.000	3.753	1.8	3.0	−6.2
VPA		2	2.07	6.5	11.1	3.5
		3.2	3.33	4.5	9.1	3.9
		30	30.47	3.7	14.1	1.6
		160	165.40	6.4	3.3	3.4

**Table 2 molecules-28-02439-t002:** Matrix effects and recoveries for determination of VAN, MER, and VPA in serum and CSF.

Analyte	Matrix	Matrix Effects (%)			Recovery (%)		
		Low QC	Medium QC	High QC	Low QC	Medium QC	High QC
VAN	Serum	86.0 ± 3.0	76.7 ± 2.6	74.0 ± 3.6	74.6 ± 1.4	74.2 ± 1.2	72.4 ± 5.3
MER		88.5 ± 12.0	80.3 ± 3.9	86.6 ± 9.4	85.6 ± 5.2	76.6 ± 13.3	79.2 ± 7.6
VPA		78.3 ± 1.9	82.8 ± 1.6	75.1 ± 1.6	72.1 ± 3.2	76.7 ± 1.1	74.4 ± 3.0
VAN	CSF	84.7 ± 9.4	70.8 ± 4.4	74.7 ± 1.6	73.8 ± 6.7	86.1 ± 14.0	73.0 ± 4.4
MER		83.8 ± 5.1	82.7 ± 2.9	85.1 ± 2.4	95.4 ± 2.4	89.6 ± 6.1	96.8 ± 3.6
VPA		79.4 ± 8.8	75.0 ± 4.7	72.4 ± 1. 8	95.9 ± 6.2	93.8 ± 2.0	85.5 ± 2.0

**Table 3 molecules-28-02439-t003:** Stability data for VAN, MER, and VPA in human serum and CSF under various conditions (data based on assays of three samples at each concentration).

Analyte	Matrix	Nominal Concentration (ng/mL)	At RT for 6 h		In the Autosampler for 6 h		Processed Samples at RT for 6 h		Three Freeze–Thaw Cycles		At −20 °C for 30 Days	
			RSD (%)	RE (%)	RSD (%)	RE (%)	RSD (%)	RE (%)	RSD (%)	RE (%)	RSD (%)	RE (%)
VAN	Serum	160	6.3	−2.7	6.3	−1.7	6.3	0.4	6.7	−7.3	6.3	−0.4
		8000	1.7	2.0	5.0	−4.9	5.7	−3.2	2.9	4.8	1.1	−12.4
MER		8	3.6	5.2	5.4	4.0	6.7	0.8	6.6	−2.0	9.2	0.8
		400	1.2	6.6	4.6	−5.3	1.4	0.2	7.0	−12.8	5.5	0.8
VPA		1600	3.0	5.6	3.6	3.3	2.3	9.4	5.3	6.9	5.2	−4.0
		80,000	5.9	4.1	4.6	3.7	5.4	6.5	1.3	3.3	1.4	9.4
VAN	CSF	1600	3.7	1.7	4.6	8.1	3.5	6.0	3.4	10.0	7.3	2.3
		80,000	2.0	0.0	1.0	5.9	2.1	0.5	4.7	1.9	11.6	−1.6
MER		80	2.9	−10.1	5.4	3.0	0.4	9.3	6.7	4.8	2.0	−4.4
		4000	1.8	−2.3	3.4	−8.1	1.4	−5.3	2.6	−3.1	4.3	4.0
VPA		3200	3.7	2.3	2.6	7.8	1.4	11.0	2.4	6.6	8.8	−0.9
		160,000	0.8	10.3	4.2	7.5	6.7	−0.4	4.0	4.4	3.5	7.4

**Table 4 molecules-28-02439-t004:** CSAR and concentrations of VAN, MER, and VPA for selected patients.

Analyte	Concentration (μg/mL)		CSF Albumin (mg /L)	Serum Albumin (g/ L)	CSAR
	Serum	CSF			
VAN (n = 9)	2.44	6.24	1020	30.70	33.22
	3.83	11.40	1990	30.00	66.33
	1.93	12.60	1860	30.40	61.18
	4.96	7.33	750	35.00	21.43
	2.51	2.57	500	35.30	14.16
	1.78	3.61	860	36.20	23.76
	1.31	3.21	1540	36.80	41.85
	1.30	4.56	920	31.10	29.58
	1.16	7.68	1830	27.20	67.28
MER (n = 8)	1.12	2.61	990	33.30	29.73
	0.29	9.50	970	30.50	31.80
	0.92	12.10	660	27.70	23.83
	1.15	1.06	1020	30.70	33.22
	0.16	4.12	970	31.80	30.50
	0.55	7.33	1360	39.32	34.59
	0.66	1.34	750	35.00	21.43
	1.33	6.66	770	33.80	22.78
VPA (n = 14)	41.80	50.80	1460	31.80	45.91
	40.40	77.20	1330	33.00	40.30
	10.60	15.70	750	35.00	21.43
	35.50	69.60	1260	35.30	35.69
	55.50	112.00	4070	37.30	109.12
	46.30	155.00	12,690	37.00	342.97
	34.60	78.90	10,650	37.00	287.84
	30.30	23.30	1240	36.80	33.70
	25.50	51.40	1540	36.80	41.85
	12.90	62.80	920	31.10	29.58
	9.61	26.40	2410	37.20	64.78
	31.80	46.40	880	30.90	28.48
	49.70	55.50	1710	42.60	40.14
	18.10	35.20	1830	27.20	67.28

## Data Availability

The data presented in this study are available on request from the corresponding author.

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
