# Peer review of "A High-Performance Liquid Chromatography—Mass Spectrometry Method for Simultaneous Determination of Vancomycin, Meropenem, and Valproate in Patients with Post-Craniotomy Infection"

_molecules, 2023, doi:10.3390/molecules28062439_

Round 1

Reviewer 1 Report

The manuscript “A high-performance liquid chromatography−mass spectrometry method for simultaneous determination of vancomycin, meropenem, and valproate in patients with post-craniotomy infection” describes the methodological protocol for the determination of three drugs in human serum and CSF. Although the work seems to fit its purpose, it requires major revision. In my opinion, the optimization of the sample preparation step could be more thoroughly examined, and additional parameters should be evaluated. Thus, I recommend a major revision. Please find my comments below.

Section 2.1. Authors have checked the effect of ammonium acetate and formic acid (as additives to the mobile phase) on analyte signals. Could acetic acid be more suitable for determinations of  VAN and MER by LC-MS/MS? The same question arises in the case of section 2.2, and the adjustment of conditions of protein precipitation.

Section 2.2. During the optimization of the sample preparation step evaluating some parameters like extraction recoveries, matrix effect, or apparent recovery are useful to confirm the correctness of the proposed methodology. Did other kinds of organic solvents were tested for protein precipitation? These important issues are missing. I encourage Authors to supplement this section with additional data.

Figure 1 should be supplemented with the example of LC-MS/MS data acquired for blank CSF and serum samples.

Lines 106-107 should be supplemented with the conclusion about the strength of matrix effect in evaluated matrixes (e.g. low, negligible, strong).

Line 113: “…the stability of MER in serum and plasma differed at RT..” Plasma or CSF? Please check.

Reviewer 2 Report

The manuscript presents the development and validation of LC-MS/MS method that may be used for simultaneous determination of vancomycin, meropenem, and valproate in human serum and cerebrospinal fluid. The study results may have practical application as they can be useful for dosage optimization of the studied drugs in the clinical settings. However, there are several issues in the manuscript that need clarification.

Introduction

It is not clear from this section whether methods describing simultaneous measurement of all three drugs are available or not.

Parameters of method  validation (lines 48-55) should be shown and compared with those of the current method in the discussion section.

Methods

It was not explained how the values of  declustering potentials and collision energies were obtained.

The solvent used for dilution to obtain standard solutions was not indicated (section 3.4)?

According to the FDA guideline “A & P should be established with at least three independent A& P runs, four QC levels per run (LLOQ, L, M, H QC), and ≥ five replicates per QC level.” The Authors used three QC levels and three replicates on each day.

Line 149 - 6.1-8 10% B. (min missing?)

According to FDA guidance evaluation of matrix effect “involves comparing calibration curves in multiple sources of the biological matrix against a calibration curve in the matrix for parallelism (serial dilution of incurred samples) and nonspecific binding.” The Authors compared the mean peak areas of post-extraction blank serum and CSF samples spiked at QC concentrations with those of corresponding standard solutions to asses this effect.

Stock solution stability studies are missing.

Autosampler temperature is not provided.

Method sensitivity and the way of assessment of this parameter were not described.

The calibration curves were linear over the range 2–100 μg/mL (line 88-89). Different ranges are presented in the section 3.4 (5, 10, 30, 50, 100, 300, and 500 ng/mL for MER; 0.1, 0.2, 0.6, 1, 2, 6, and 10 μg/mL for VAN; and 1, 2, 6, 10, 20, 60, and 100 μg/mL for VPA).

The doses administered should be given in the section 3.7.

Results and discussion

Table 1- The concentration units of VPA and VAN are different than in other parts of the manuscript.

Table 4 - Meropenem concentrations should be expressed probably in different units (ng/ml). The values in the table are beyond the linear range of the calibration curve (5-500 ng/ml for serum and 50-5000 ng/ml for CSF).

The Authors should discuss and compare their results concerning the method and its validation parameters with those of other authors.

The effective concentrations of MER are 4-10 mcg/ml, whereas calibration curves  and QC samples are up to 5000 ng/ml.

The Authors should explain “how monitoring of these drugs in patients’ serum and CSF could be used to promptly adjust the dosages” in the Results and discussion section.

Reviewer 3 Report

This is a study focusing on simultaneously antibiotics concentration determination according to patients’ serum and CSF concentrations by a high-performance liquid chromatographymass spectrometry method.

The authors aimed to determine muti-antibiotic concentrations simultaneously on LC–MS/MS. But the reason why does this test simultaneously should be listed in the introduction part.

All the tests are based on authors own methods. To compared with other traditional clinical methods, rare information is indicated. I recommend author to make a comparison among this and precious methods.

The merits and demerits about your method are better to list in your discussion section.

In your method section, the flow chart on how your LC-MS/MS method performed is recommend to supply.

The D6 in your manuscript refers to?

Reviewer 4 Report

Molecules

Manuscript ID: molecules-2128657

Title: A high-performance liquid chromatography-mass spectrometry method for simultaneous determination of vancomycin, meropenem, and valproate in patients with post-craniotomy infection

Comments to the Author  

In the study, the authors aim to simultaneous determination of three antibiotic drugs including vancomycin, meropenem, and valproate which are widely used for treatment of central nervous system infection using LC-MS/MS. Overall, the manuscript is well written. The results from this study are informative and applicable for clinical application and therapeutic drug monitoring. Therefore, I am thinking that it can be acceptable for publication after revision since I have some points in the manuscript that should be clarified.

1.)     The LLOQ of vancomycin and valproate in plasma was not correlate well with the concentration range of the calibration curve. Please clarified.

2.)     Please check the concentration unit of QC sample shown in Table 2.

3.)     Please explain the criteria that the authors used for assign the concentration of plasma QC sample of vancomycin and valproate because the selected three QC concentration shown in the manuscript could not cover the low, medium and high concentration regarding to the calibration range.

4.)     Line 194; Should add reference.

5.)     Table 4: how to quantitate the level of meropenem in the plasma and CSF sample when it was over 0.5 µg/ml and 5 µg/ml, respectively.

Reviewer 5 Report

In the submitted manuscript, the authors investigated a high-performance liquid chromatography−mass spectrometry method for simultaneous determination of vancomycin, meropenem, and valproate in patients with post-craniotomy infection. Even though the article has certain scientific significance, some parts of it are incomplete. I have the following comments.

 Abstract

- The abstract should be revised and restructured.

- The results of the clinical application of the study should be included in the abstract.

 Introduction:

 - The literature review of the article on the detection of VAN, MER, and VPA levels in human serum and cerebrospinal fluid is very poor and should be revised and completed.

- The objectives of the study should be explained more precisely.

 Results and discussion

 - The result and discussion section in this paper is very important and the authors should present and explain the results obtained in their study well and compare it with the results of others. Therefore, this part should be strengthened.

 Materials and methods

- Line 164: Please write the specifications and preparation of blank human serum and CSF.

Section 3. The references used for sections 3.4, 3.5, 3.6, and 3.7 should be provided.

- Line 218: Please provide the reference code of research ethics.

 - The conclusions should be integrated with more detailed results summarizing all the study and must reflect the innovation of this study and the perspectives.

Round 2

Reviewer 1 Report

Thank you for addressing my comments. I have no additional questions.

Author Response

Thank you for reviewing the manuscript and giving us the chance to revise it. We greatly appreciate the effort in handling and reviewing our manuscript as well as the valuable comments provided.

Reviewer 2 Report

The Authors addressed most of my concerns, however, there are still several issues that need to be clarified:

The sentence: “The LLOQs of this method were 5 ng/mL for MER, 0.1 μg/mL for VAN, and 1 μg/mL for VPA in serum and 50 ng/mL for MER, 1 μg/mL for VAN, and 2 μg/mL for VPA in CSF” was repeated twice in the manuscript. It is not necessary in the Introduction section.

It is not clear why/what for the Authors presented values from Table 2 in the text after the description of method optimization in section 2.2. (lines 105-113).

The paragraph:

“In this study, due to administrate via intrathecal injection, the concentrations of these three drugs in CSF were higher than in serum. Although the CSF levels did not reach …..
of these three drugs in CSF administrated by intrathecal injection is very essential”

is repeated twice in the manuscript (lines 171-191 and 313-322).

The manuscript needs a copy editing by a native speaker of English.

Author Response

Point 1: The sentence: “The LLOQs of this method were 5 ng/mL for MER, 0.1 μg/mL for VAN, and 1 μg/mL for VPA in serum and 50 ng/mL for MER, 1 μg/mL for VAN, and 2 μg/mL for VPA in CSF” was repeated twice in the manuscript. It is not necessary in the Introduction section.

Response 1: Thank you for pointing this out. According to the Reviewer’s suggestion, this sentence in the Introduction section is removed.

Point 2: It is not clear why/what for the Authors presented values from Table 2 in the text after the description of method optimization in section 2.2. (lines 105-113).

Response 2: Thank you for pointing this out. According to the Reviewer’s suggestion, the values from Table 2 presented in Section 2.2 is removed.

Point 3: The paragraph:“In this study, due to administrate via intrathecal injection, the concentrations of these three drugs in CSF were higher than in serum. Although the CSF levels did not reach …..of these three drugs in CSF administrated by intrathecal injection is very essential”is repeated twice in the manuscript (lines 171-191 and 313-322).

Response 3: Thank you for the suggestion. The paragraph (lines 313-322) is polished as follow: “Although the CSF levels in this study did not reach the effective concentration reported, these three drugs still provide sufficient therapeutic effects. It indicated the effective concentrations of these three drugs in CSF administrated by intrathecal injection maybe lower than administrated via intravenous injection. It is very essential to investigate the effective concentrations of these three drugs in CSF administrated by intrathecal injec-tion. Our method provided a detection tool to investigate the effective concentrations of these three drugs in CSF from patients administered via intrathecal injection. For this reason, our method is an effective analysis approach for clinical purposes.”

Point 4: The manuscript needs a copy editing by a native speaker of English.

Response 4: Thank you for pointing this out. The manuscript has been polished by the Mogo Internet Technology Co., a professional editing company. And the CERTIFICATE OF ENGLISH EDITING could be provided.

Reviewer 4 Report

Dear Authors,

Thank you for your revised version. I appreciate your detailed response to the reviewer.  In principle, I am therefore inclined to accept your revised version for publication.

Author Response

(The authors gave the same response as above.)

Reviewer 5 Report

-

Author Response

Thank you for reviewing the manuscript and giving us the chance to revise it. We greatly appreciate the effort in handling and reviewing our manuscript as well as the valuable comments provided.  And the results are improved.